# Comprehensive Ocular and Systemic Safety Evaluation of Polysialic Acid-Decorated Immune Modulating Therapeutic Nanoparticles (PolySia-NPs) to Support Entry into First-in-Human Clinical Trials

**DOI:** 10.3390/ph17040481

**Published:** 2024-04-09

**Authors:** Anitha Krishnan, David G. Callanan, Victor G. Sendra, Amit Lad, Sunny Christian, Ravinder Earla, Ali Khanehzar, Andrew J. Tolentino, Valory Anne Sarmiento Vailoces, Michelle K. Greene, Christopher J. Scott, Derek Y. Kunimoto, Tarek S. Hassan, Mohamed A. Genead, Michael J. Tolentino

**Affiliations:** 1Aviceda Therapeutics, Cambridge, MA 02142, USA; akrishnan@avicedarx.com (A.K.); dcallanan@avicedarx.com (D.G.C.); alad@avicedarx.com (A.L.); schristian@avicedarx.com (S.C.); rearla@avicedarx.com (R.E.); akhanehzar@avicedarx.com (A.K.); mgreene@avicedarx.com (M.K.G.); cscott@avicedarx.com (C.J.S.); dkunimoto@avicedarx.com (D.Y.K.); thassan@avicedarx.com (T.S.H.); mgenead@avicedarx.com (M.A.G.); 2Department of Biology, University of California Berkeley, Berkeley, CA 94720, USA; atolent@berkeley.edu; 3School of Medicine and Health Sciences, George Washington University, Washington, DC 20052, USA; valory_vailoces@gwu.edu; 4The Patrick G. Johnston Centre for Cancer Research, School of Medicine, Dentistry & Biomedical Sciences, Queen’s University Belfast, Belfast BT9 7AE, UK; 5Oakland University William Beaumont School of Medicine, Royal Oaks, MI 48067, USA; 6Department of Ophthalmology, University of Central Florida School of Medicine, Orlando, FL 32827, USA; 7Department of Ophthalmology, Orlando College of Osteopathic Medicine, Orlando, FL 34787, USA

**Keywords:** nanoparticle, polysialic acid, toxicity, macular degeneration, geographic atrophy, intravitreal administration, intravenous administration

## Abstract

An inflammation-resolving polysialic acid-decorated PLGA nanoparticle (PolySia-NP) has been developed to treat geographic atrophy/age-related macular degeneration and other conditions caused by macrophage and complement over-activation. While PolySia-NPs have demonstrated pre-clinical efficacy, this study evaluated its systemic and intraocular safety. PolySia-NPs were evaluated in vitro for mutagenic activity using *Salmonella* strains and *E. coli*, with and without metabolic activation; cytotoxicity was evaluated based on its interference with normal mitosis. PolySia-NPs were administered intravenously in CD-1 mice and Sprague Dawley rats and assessed for survival and toxicity. Intravitreal (IVT) administration in Dutch Belted rabbits and non-human primates was assessed for ocular or systemic toxicity. In vitro results indicate that PolySia-NPs did not induce mutagenicity or cytotoxicity. Intravenous administration did not show clastogenic activity, effects on survival, or toxicity. A single intravitreal (IVT) injection and two elevated repeat IVT doses of PolySia-NPs separated by 7 days in rabbits showed no signs of systemic or ocular toxicity. A single IVT inoculation of PolySia-NPs in non-human primates demonstrated no adverse clinical or ophthalmological effects. The demonstration of systemic and ocular safety of PolySia-NPs supports its advancement into human clinical trials as a promising therapeutic approach for systemic and retinal degenerative diseases caused by chronic immune activation.

## 1. Introduction

Pegcetacoplan, a c3 depletion agent, and avacincaptad pegol, a c5 depletion agent, are currently FDA-approved therapeutics for geographic atrophy secondary to age-related macular degeneration that modestly reduce the rate of growth of geographic atrophy when injected intravitreally once a month [1,2]. The limitations of both of these treatments are the absence of functional visual benefit coupled with associated serious ocular toxicity, including choroidal neovascular conversion, retinal vasculitis, and optic neuritis [3]. A polysialic acid-decorated nanoparticle (PolySia-NP) developed to target Siglecs 7, 9, and 11 and activate complement factor H (CFH) to repolarize activated macrophages/microglia to the resolution state and abrogate the amplification of the alternative complement pathway was designed to both halt the growth of geographic atrophy and recover visual function by increasing the population of neuroprotective growth factor-producing microglia/macrophages that rescue pre-apoptotic photoreceptors [4,5]. In vitro and in vivo evaluation of this PolySia-NP demonstrated the ability to repolarize phagocytic, inflammatory, fibrotic, and neovascular polarized macrophage/microglial states to the resolution/healing state [5].

The promise of obtaining visual gain in patients with geographic atrophy has progressed PolySia-NPs into human clinical trials for the treatment of geographic atrophy secondary to age-related macular degeneration (GA/AMD) [5,6]. PolySia-NPs represent the first therapeutic sialic acid-coated self-associated molecular pattern (saSAMP) mimetic nanoparticle to enter human trials and has the potential to be a best-in-class therapeutic for GA/AMD.

The use of disialic acid-decorated polylactic-co-glycolic acid (PLGA) nanoparticle self-associated molecular pattern mimetics that agonize sialic acid-binding Ig-like lectins (SIGLECs) to abrogate adaptive and innate immune-mediated inflammation was first demonstrated in mouse models of induced sepsis [7]. The potent anti-inflammatory properties of sialic acid-coated nanoparticles result from immune synapse formation with inflammation-resolving SIGLEC receptors on the surface of activated immune cells. Diverse presentations of end-linked sialic acid glycan patterns on the cellular glycome of healthy host cells are the main markers of immune self and have been described as self-associated molecular patterns (SAMPs) [8]. SIGLEC receptors, which bind these sialic acid patterns, are the main checkpoint receptors that prevent autoimmunity and are responsible for both adaptive and innate inflammatory resolution. SIGLECs are considered key immune-resolving receptors because they represent the largest family of checkpoint receptors that contain the immunoreceptor tyrosine-based inhibitory motif (ITIM), a cytoplasmic domain that, when activated, recruits powerful tyrosine protein phosphatases called SHP-1/2. These locally recruited phosphatases dephosphorylate inflammatory cell activation pathways, resulting in a profound resolution of immune cell activation [9].

The most well-studied checkpoint receptor that contains the ITIM domain is the programmed death-1 receptor (PD-1) [10]. PD-1 recognizes the protein PD-1 ligand, which is found in limited distribution and does not represent a significant marker of self. In contrast, SIGLECs bind to end-linked sialic acid-self-associated molecular patterns that healthy host cells use as the main identifier of self. The PolySia-NP, which was designed to agonize the SIGLECs found on macrophages and microglial cells, is a nanoparticle with a PLGA core decorated with polysialic acids that are conjugated to the core with polyethylene glycol (PEG) linkers [5]. PLGA in the form of a dexamethasone-releasing pellet has been FDA-approved and found safe for intravitreal use in humans [11]. PEG as a component of other intravitreal drugs has been evaluated and found safe [12,13]. While PolySia has not been tested for safety, it is a naturally occurring sialic acid moiety and is found in abundance in the central nervous system [14]. While components of PolySia-NPs are likely safe, the development of PolySia-NPs as the actual drug substance requires in vitro and in vivo safety testing before administration to humans.

The use of nanoparticles in pharmaceuticals has predominantly focused on delivering active drug substances [15]. Nanoparticle-based technologies offer exciting new approaches to disease diagnostics and therapeutics, as they can enter the tissues at the molecular level. Nanoparticle-based formulations offer effective cellular uptake, reduce rapid clearance, and provide a targeted delivery approach. Nanoparticles are 1–100 nanometers in size. These particles can be used in a targeted drug delivery approach, as they can specifically target a drug or drug carrier to minimize drug-originated systemic toxic effects. In addition, nanoparticles are designed to assist therapeutic agents in passing through biologic barriers, mediate molecular interactions, and identify molecular changes. The different types of nanoparticle-based formulations are polymeric nanoparticles, liposomes, drug-conjugated nanoparticles, metal nanoparticles, polymeric micelles, and many more, which have been reported in various studies. Active pharmaceutical ingredients (APIs) can be entrapped, encapsulated, dissolved, or linked to the nanoparticle matrix [16,17]. Nanoparticles for ophthalmic applications provide a new platform for topical delivery or carrying small molecules for the treatment of ocular diseases. The development of nanoformulations aims to possess good biocompatibility with the eye, overcome ocular barriers, and maintain effectiveness over time in the targeted ocular tissues [4,5,18,19].

PolySia-NPs represent the first instance of a glycan-coated nanoparticle as the actual drug substance, representing a novel class of immune-modulating therapeutics [20,21]. While components of the nanoparticle have proven safe and nontoxic, evaluation of the toxicity of novel glycomimetic nanoparticles is unknown. The purpose of this study is to evaluate the in vitro mutagenicity and cytotoxicity, in addition to in vivo ocular and systemic toxicity and clastogenicity, for clinically relevant doses of PolySia-NPs to enter this first-in-class therapeutic for geographic atrophy/macular degeneration into human clinical trials.

## 2. Results

### 2.1. PolySia-NPs Did Not Show Mutagenic Activity

Precipitates were not observed in any strain either with or without metabolic activation for PolySia-NPs at 1.0, 5.0, 10, 50, 100, 500, 1000, and 5000 µg/plate. Cytotoxicity (i.e., reduction in the background lawn and/or number of revertant colonies) was not observed in any strain with or without metabolic activation (Table 1).

Similarly, precipitates were not observed in any strain either with or without metabolic activation in the mutagenicity assay at 31, 93, 278, 833, and 2500 µg/plate of PolySia-NPs by using the plate incorporation method. Cytotoxicity (i.e., reduction in the background lawn and/or mean number of revertant colonies) was not observed in any strain (TA98, TA100, TA1535, TA1537, and WP2uvrA) either with or without metabolic activation (Table 2 and Table 3).

### 2.2. PolySia-NPs Did Not Show Cytotoxicity In Vitro

Precipitates were not observed in any treatment with or without metabolic activation, and the PolySia-NP concentrations tested in the range-finding assay ranged from 0.975 to 500 μg/mL.

In the 4 h treatment with metabolic activation, the concentrations selected for micronucleus analysis were as follows (with percent cytotoxicity, using RICC): 31.3 μg/mL (<0%), 62.5 μg/mL (<0%), 125 μg/mL (<0%), 250 μg/mL (<0%), and 500 μg/mL (3%) (Table 4).

Similarly, in the 4 h treatment without metabolic activation, the concentrations selected for micronucleus analysis were as follows (percent cytotoxicity, using RICC): 31.3 μg/mL (<0%), 62.5 μg/mL (<0%), 125 μg/mL (<0%), 250 μg/mL (<0%), and 500 μg/mL (<0%) (Table 5).

In the 27 h treatment without metabolic activation, the concentrations selected for micronucleus analysis were as follows (with percent cytotoxicity, using RICC): 31.3 μg/mL (<0%), 62.5 μg/mL (<0%), 125 μg/mL (<0%), 250 μg/mL (3%), and 500 μg/mL (1%). (Table 6).

No statistically significant test article-related increase in micronucleated cells was determined in all three conditions. The formulation cultures, along with the vehicle and two concentrations of positive control for each treatment condition, were analyzed for the presence of micronuclei. Micronuclei were evaluated in approximately 10,000 cells per culture. No statistically significant increases in the percent of micronucleated cells were noted between formulation-treated cultures and the concurrent vehicle control under any assay condition. Therefore, PolySia-NPs were considered negative for inducing micronuclei in TK6 cells in the 27 h treatment without metabolic activation and negative for inducing micronuclei in TK6 cells in the 4 h treatments with and without metabolic activation.

### 2.3. PolySia-NPs Did Not Show Micronucleus Formation In Vivo after Intravenous (IV) Administration

Animals received dose levels of 0, 208.75, 417.5, and 835 mg/kg/day from groups 1 to 4 to determine the high dose for the definitive micronucleus assay. After dosing, no notable abnormal clinical signs or body weight change were observed in the test article treatment group animals. Considering a maximum feasible concentration of 33.4 mg/mL and a maximum IV injection volume of 25 mL/kg, 835 mg/kg/day was selected as the high dose level for the definitive assay.

After dosing, all dosed animals survived to the scheduled termination. Abnormal clinical signs of skin discoloration (tail, middle) were found in the 417.5 and 835 mg/kg/day dose groups; no other abnormal sign was found in any other groups (Appendix A).

On Day 3, compared with Day 1, no obvious body weight reduction was found in any treatment groups (Appendix A). When compared with the negative control group, no obvious reductions in the percentage of PCE among total erythrocytes were observed in any treatment groups.

No statistically significant increases in micronucleus formation (*p* > 0.05) were observed in any test article dose groups when compared with the control article group. No significant dose-related increase response was identified by statistical analysis for test article treatment groups (Cochran–Armitage, *p* > 0.05) (Table 7). The micronucleus frequencies of all PolySia-NPs were within the 99% control limits of the laboratory’s historical negative control data range. The positive control article induced the expected statistically significant increase in micronucleus formation (*p* ≤ 0.01) when compared to the concurrent control article group.

### 2.4. Intravenous (IV) Administration of PolySia-NPs Did Not Show Signs of Toxicity in Mice and Rats

Following a once weekly IV injection of PolySia-NPs at 1.6, 8, or 16 mg/kg/dose to male mice for 5 doses followed by a 14-day recovery period, no moribidity or death was reported. There were no changes in clinical observation, body weight/body weight gain (Figure 1A), food consumption (Figure 1B), ophthalmology, clinical pathology, organ weight, or macroscopic and microscopic observation noted.

The evaluation of PolySia-NPs on respiratory functions when administered as a single dose by intravenous injection (IV) in rats did not show any deaths noted during this study. There were no changes in all respiratory system functional parameters (tidal volume, derived minute volume, and respiratoration rate) up to 24 h following a single intravenous administration in Sprague Dawley rats (Figure 2A, B, and C, respectively).

### 2.5. Intravitreal (IVT) Administration of PolySia-NPs Was Well-Tolerated in Rabbits

A single administration of PolySia-NPs by intravitreal injection at doses of 0.050, 0.150, and 0.500 mg/eye to Dutch Belted rabbits was well tolerated on Days 29 or 43. There were no mortalities and no PolySia-NP-related clinical observations or effects on body weights, body weight changes (Figure 3), food consumption, hematology, coagulation, or clinical chemistry parameters.

There were no gross findings, organ weight changes, or systemic microscopic findings considered related to the intravitreal administration of PolySia-NPs. Microscopic changes considered possibly related to PolySia-NPs were present in the eyes at ≥0.15 mg/eye on Days 29 and 43 (Table 8).

Minimal to mild mononuclear cell infiltration was observed unilaterally or bilaterally in the vitreous chamber of most animals of both genders given PolySia-NPs at ≥0.15 mg/eye, with a dose-related increasing trend in severity and no clear evidence of reversibility on Day 43. The severity of mononuclear cell infiltration at 0.15 mg/eye was restricted to minimal, while it ranged from minimal to mild at 0.500 mg/eye. This microscopic change correlated with vitreal-likeopacities (cell type) at ophthalmology examination. However, mononuclear cell infiltration can occur following intravitreal administration of endotoxins. Therefore, the endotoxin levels reported in the test item construct at 0.15 and 0.5 mg/eye (endotoxin levels of 0.022 and 0.075 EU/50 μL, respectively) may or may not have contributed to the observed microscopic findings under the conditions of this study.

In another study, the left eyes (OSs) were IVT injected with ultra-high-dose PolySia-NPs (40 mg/mL), and the right eyes (ODs) were IVT injected with high-dose PolySia-NPs (10 mg/mL). Baseline ocular examinations (OEs) were normal in all eyes, but on Day 2, the OSs had mild or moderate inflammation while the ODs were not inflamed. On Day 4, the OSs had moderate or severe inflammation, while the ODs had mild or no inflammation (Figure 4A). On Day 7, all eyes had trace inflammation without any topical or systemic treatment. Similarly, at 20 mg/mL, mild inflammation was observed at Day 7, with improvement at Day 14 (Figure 4B).

Optical coherence tomography (OCT) was performed on Day 7 to measure ONL and total retinal thickness. The outer nuclear layer (ONL) thickness was measured, and the OSs (PolySia-NP 40 mg/mL) had a slightly larger average thickness than the ODs (PolySia-NP 10 mg/mL), at 57.6 ± 2.6 μm compared to 49.1 ± 5.1 μm (Figure 5A). Total retinal thickness was slightly larger in the OSs than in the ODs on Day 7 (185.3 ± 4.9 μm vs. 171.9 ± 11.0 μm (Figure 5B), within the total thickness baseline range of 179–187 μm recorded previously (Krishnan et al., 2023) [5]. Based on the results, 20 mg/mL was chosen as a tolerable dose of PolySia-NPs.

### 2.6. Intravitreal (IVT) Administration of PolySia-NPs Was Well-Tolerated in Non-Human Primates (NHPs)

As primates have a close Siglec homology to humans, safety studies in primates are required by regulatory health authoritieso obtain a better understanding of the potential safety of any new molecules to be tested in humans. We performed a safety study in non-human primates (Cynomolgus macaque) monkeys, demonstrating the safety of IVT inoculation for up to 28 days, followed by 14 days of recovery. Overall, there were no clinical observations of body weight changes in male and female monkeys (Figure 6B,C and Appendix A). The individual monkey body weights generally fluctuated throughout the study with no consistent trends or dose response and, therefore, were attributed to individual animal variability, stress response, or clinical condition (e.g., fecal changes) or considered procedure-related. Additionally, there were no organ weight changes, microscopic and macroscopic findings, or definitive effects related to IVT inoculation of PolySia-NPs.

The IVT inoculation of PolySia-NPs at a single dose of 0.050, 0.150, and 0.500 mg/eye or vehicle to monkeys resulted in no ophthalmoscopic adverse effects (indirect, slit-lamp) with no alterations in retinal functions detected on the intraocular pressure (IOP) and full-field ERG readings (Figure 7A,B) at the terminal or recovery phase compared to baseline (pre-test). Retinal abnormalities were not detected on OCT images for all the groups (Figure 7C).

Clinically, we detected several monkeys exhibiting a minimal to moderate self-limiting inflammatory response in the eye at high doses, causing a slight hazy appearance within the vitreous that resolved completely within days and was considered well tolerated without sequelae during the study. Based on these results, the No-Observed-Adverse-Effect-Level (NOAEL) was 0.500 mg/eye from a single intravitreal injection over a 29-day (males) or 28-day (females) observation period.

## 3. Discussion

Sialic acid-decorated SAMP mimetic nanoparticles have the potential to become potent immune cell-modulating therapeutics that can be used to treat many diverse inflammatory diseases with unmet medical needs. Demonstrating the safety and lack of toxicity of PolySia-NPs after ocular and systemic administration paves the way for the development not only of PolySia-NPs but other saSAMP mimetic nanoparticle therapeutics.

The development of a non-inflammatory, non-pro-angiogenic, geographic atrophy halting and vision maintenance therapeutic for geographic atrophy secondary to age-related macular degeneration is critical due to the lack of visual function effects and clinical side effects of recently approved therapies for GA/AMD. Pegcetacoplan and avacincaptad pegol have demonstrated the potential to induce retinal vasculitis, optic neuritis, and conversion to the exudative neovascular form of AMD, but are not able to improve BCVA or significantly halt the progression of GA. While PolySia-NPs downmodulate both microglial/macrophage polarization and alternative complement pathways, their mechanism does not involve depletion of complement factors or cytokines [5]. Therefore, PolySia-NPs should be safer to use in an in vivo environment [20]. PolySia-NPs have been shown to decrease microglial/macrophage VEGF production and were able to inhibit choroidal neovascularization in a laser-induced model of exudative AMD at the same potency as aflibercept, a clinically utilized anti-VEGF intravitreal injection [4]. While able to prevent amplification of the alternative complement pathway commensurate to pegcetacoplan and avacincaptad pegol, their anti-angiogenic properties will prevent the conversion of non-exudative AMD to the exudative form, which appears as a significant adverse event in patients receiving the approved therapies for GA/AMD [1,2].

Genotoxicity has been attributed to titanium dioxide and silver nanoparticles (TiO_2_ and AgNPs) [22], but was not seen in the poly-lactic-co-glycolic acid-polyethylene oxide copolymer (PLGA-PEO) [23]. PolySia-NPs demonstrate no genotoxicity in the in vitro bacterial reverse mutation assay, consistent with the PLGA-PEO nanoparticle. In line with PLGA-PEO, PolySia-NPs containing PLGA demonstrated no evidence of mitogenic inhibition or clastogenicity, which may be specific to metal-derived nanoparticles [22]. From these in vitro studies, PolySia does not enhance the genotoxicity, mitogenic inhibition, or clastogenicity of the PLGA nanoparticle.

Previous evaluation of the in vivo clastogenicity of a PLGA nanoparticle was in the setting of antitubercular drug-loaded PLGA nanoparticles [24]. The PLGA nanoformulation of levofloxacin and ethionamide showed less clastogenicity in Swiss albino mice than levofloxacin or ethionamide alone [24], possibly indicating that the PLGA nanoparticle reduced clastogenicity. Systemic administration of PolySia-NPs containing PLGA demonstrated no evidence of clastogenicity, in line with the reduction of clastogenicity attributed to PLGA nanoparticles.

Systemic administration of PolySia-NPs was also found to produce no clinical signs of toxicity, consistent with what has been seen when PLGA nanoparticles have been administered systemically [25,26]. The previous formulations of in vivo-tested PLGA nanoparticles were tested as carriers of pharmaceutical agents like Gliclazide [26] and ant-tubercular drugs [25]. PolySia-NPs are the therapeutic drug substance and is not used to reduce toxicity but to agonize Siglec receptors and CFH. The hypothesis that PolySia-NPs would prove safe since they are not used to deliver toxic reagents was confirmed in its in vivo evaluation for systemic toxicity, demonstrating pre-clinical safety.

The evaluation of PolySia-NPs for intravitreal administration has also been evaluated in the setting of drug delivery to the retina [27,28,29,30]. Fenofibrate (peroxisome proliferator-activated receptor alpha (PPAR) agonist)-encapsulated PLGA nanoparticles and blank PLGA NP were tested intravitreally in streptozocin-induced diabetic rats, laser-induced choroidal neovascularization, and very low-density lipoprotein receptor knockout (Vldlr −/−) mice, and no toxicity to retinal structure and function was noted [31]. PolySia-NPs demonstrated no functional or ophthalmic toxicity. PolySia-NPs are also potent anti-inflammatory compounds, and intravitreal injections do not result in inflammatory responses, potentially because PolySia-NPs are not pro-inflammatory in the eye or because of their potent anti-inflammatory properties [4,5]. There was also no effect on retinal function as measured by electroretinogram, like the negative findings when fenofibrate-containing nanoparticles were injected intravitreally and eyes were examined by ERG [31]. Intravitreal PLGA has been shown pre-clinically in nanoparticles and clinically in a PLGA-dexamethasone eluting implant to be safe and non-toxic [29,32].

While PolySia has not been tested for safety after intravitreal administration, PolySia is ubiquitous in the central nervous system, and this study does not appear to be toxic when presented on the surface of a PLGA nanoparticle. PolySia was found to be neuroprotective to the retinal ganglion after kainic acid-induced excitotoxicity [33]. Intravitreal administration of PolySia was demonstrated to inhibit macrophage recruitment, complement activity, and neovascularization in a laser-induced model of choroidal neovascularization [34,35]. No toxicity study of intravitreal PolySia has been performed, but the endogenous presence of PolySia in the retina and CNS and its therapeutic effect in animal models support the results of this paper, which show no ocular toxicity towards a PolySia-NP.

Based on the safety demonstrated in this paper, intravitreally administered PolySia-NPs have entered human clinical trials for the treatment of geographic atrophy secondary to age-related macular degeneration. The safety of PolySia-NPs in humans has been demonstrated in a dose-escalating single injection phase II study and will be further examined in a randomized, multi-dose follow-up study (SIGLEC study ClinicalTrials.gov Identifier NCT05839041).

## 4. Materials and Methods

### 4.1. PolySia-NP Preparation Method

PolySia-NPs are manufactured by first covalently conjugating PLGA-PEG polymer and PSA (polysialic acid). A polymer phase is prepared by dissolving PLGA-PEG polymer in a suitable vehicle. Separately, the PSA phase is prepared, where PSA is dissolved in a suitable vehicle. Both phases are mixed and kept for conjugation for a pre-defined time. This is identified as the organic phase during manufacturing. Separately, an aqueous phase is prepared. For nanoparticle formation, a primary coarse suspension is formed by mixing the organic phase with the aqueous phase. The suspension is further processed via a high-pressure homogenizer to reduce the final average particle size to ~100 nm. The resulting particles are purified using a filtration process to remove excess solvents and unwanted impurities. The purified suspension is then subjected to tonicity and pH adjustment, pre-filtration, followed by filtration through a 0.2-micron filter under laminar flow.

### 4.2. Bacterial Reverse Mutation Assay

This study was performed in accordance with Good Laboratory Practice, U.S. Department of Health and Human Services, Food and Drug Administration. PolySia-NPs were evaluated for mutagenic activity in an in vitro bacterial reverse mutation assay. Four tester strains of *Salmonella typhimurium* (TA98, TA100, TA1535, and TA1537) and 1 Escherichia coli strain (WP2 uvrA) were used for mutagenicity testing [36]. Mutagenicity testing was performed in triplicate at each concentration with and without a phenobarbital/5,6-benzoflavone-induced rat liver S9 metabolic activation system. PolySia-NPs were tested at 1.0, 5.0, 10, 50, 100, 500, 1000, and 5000 µg/plate using the plate incorporation method. The vehicle was used as the control.

### 4.3. In Vitro Micronucleus Assay

PolySia-NPs were evaluated for the potential to induce micronuclei in TK6 cells during short (4 h) and long (27 h) incubations with or without an exogenous metabolic activation system. The in vitro micronucleus assay evaluates the ability of the formulations to interfere with normal mitotic cell division [37]. Micronuclei represent damage transmitted to daughter cells and originate from acentric fragments or whole chromosomes that do not migrate to the poles during anaphase. At telophase, these chromosomes and/or fragments are not segregated to either the daughter nucleus and form a single micronucleus or multiple micronuclei in the cytoplasm. A micronucleus assay is used as an appropriate test for detecting potential clastogens and aneugens in cells after exposure to the formulations [38]. An extended (40 h) recovery was used in this assay, which has been shown to optimize the micronucleus response after treatment with some positive substances when compared to a standard (24 h) recovery [39].

TK6 cultures were treated with PolySia-NPs (0.975 to 500 μg/mL), positive control, or vehicle control in the presence and absence of phenobarbital/5,6-benzoflavone-induced rat liver S9 microsomal fraction. The 10% sucrose concentration in the culture medium was 2.6% (*v*/*v*). The positive controls, mitomycin C (MMC, 0.0625 and 0.125 μg/mL; Sigma-Aldrich, St. Louis, MO, USA) for the 4 h treatment without metabolic activation, vinblastine sulfate (VIN, 2.5 and 3.0 ng/mL; Sigma-Aldrich) for the 27 h treatment without metabolic activation, and cyclophosphamide monohydrate (CP, 4.7 and 11.9 μg/mL; Sigma-Aldrich) for the 4 h treatment with metabolic activation, were chosen as metabolism-dependent (CP) and direct-acting (MMC and VIN) positive controls. All cultures were examined visually for signs of cytotoxicity (i.e., visible cell abnormalities), pH change, and precipitates at the time of dosing, at the end of the 4 h treatment (at wash), and before harvest.

Cultures selected for micronucleus evaluation were processed according to the in vitro MicroFlow kit (Litron Labs, Rochester, NY, USA) and analyzed by flow cytometry using a flow cytometer with BD FACSDiva software (https://www.bdbiosciences.com/en-us/products/software/instrument-software/bd-facsdiva-software). Positive control and cultures treated with PolySia-NPs were compared to the corresponding vehicle control cultures using *z*’ [40]. As a general guideline, the tested PolySia-NPs were clearly considered positive for inducing micronuclei if: 1. a significant increase (z′ ≥ 0.6) in the percentage of cells with micronuclei is observed at 1 or more concentrations; 2. the increase is dose-related in at least 1 experimental condition when evaluated with an appropriate trend test; and 3. at least 1 concentration induces an increase in % MN that is above the upper bounds of the negative historical control distribution. The test article was clearly considered negative for micronuclei induction if all the positive response criteria were unmet.

The in vivo studies were performed in accordance with the OECD Principles of Good Laboratory Practice and as accepted by regulatory authorities throughout the European Union, the United States of America (FDA), Japan (MHLW), and other countries that are signatories to the OECD Mutual Acceptance of Data Agreement.

### 4.4. Clastogenic Activity Assessment after Administration by Intravenous (IV) on Mice

This study was performed to determine whether PolySia-NPs exhibit clastogenic activity and/or disrupts the mitotic apparatus by evaluating its ability to induce an increase in micronuclei frequency in polychromatic erythrocyte cells (PCE) in the bone marrow of adult male CD-1 mice after administration by intravenous (IV) injection. Twenty-seven animals were assigned to 5 groups, with 5 animals per group (7 animals for the high dose group) and treated with a vehicle/negative control article (Group 1), PolySia-NPs (Groups 2 to 4), or a positive control (Group 5). The dose levels in Groups 2, 3, and 4 were 208.75, 417.5, and 835 mg/kg/day, respectively. The vehicle/negative control article and PolySia-NPs were administered once daily by IV injection for 3 days to animals with a dose volume of 25 mL/kg. A total of 75 mg/kg of the positive control article (Cyclophosphamide monohydrate) was given to mice once by IP injection at a dose volume of 10 mL/kg. Bone marrow was collected from each animal in Group 5 for micronucleus frequency analysis at 22 (20–24) hours post-last dosing. All animals were clinically monitored and euthanized on Day 4 at 22 (20 to 24) hours post-last dosing treatment.

### 4.5. Toxicity Assessments of PolySia-NPs after Intravenous Administration (IV) in Mice and Rats

Crl:CD1 mice (6/group) were administered vehicle (10% Sterile Sucrose) or PolySia-NPs at 1.6, 8.0, or 16.0 mg/kg/dose once weekly for 5 doses by intravenous (IV) injection (5 mL/kg) on Days 1, 8, 15, 22, and 29. Morbidity/mortality, clinical observations, body weight, food consumption, clinical pathology (hematology and serum chemistry), and gross (necropsy) were evaluated.

In addition, the toxicity of PolySia-NPs was evaluated on respiratory system function following a single intravenous injection (IV) administration in Sprague Dawley rats. Thirty-two (32) rats (16 rats/sex), 6 to 7 weeks of age, were randomly assigned to 4 groups of 4/sex/group and administered the vehicle or PolySia-NPs in vehicle at single dose levels of 1, 3, or 10 mg/kg via intravenous injection. The dosage volume for all animals was 1 mL/kg. Prior to dosing, the respiratory data (respiratory rate, tidal volume, and minute volume) were recorded for approximately 15 min to allow assessment of baseline respiratory parameters before randomization. On the dosing day, animals were placed in ‘head-out’ plethysmograph tubes and allowed to acclimate to environmental conditions for at least 5 min prior to each data collection period. Immediately following the acclimation period, ventilatory parameters, including tidal volume, respiratory rate, and derived minute volume, were measured for 15 min once at pre-dosing and once at 6 h, 12 h, and 24 h post-dosing.

### 4.6. Toxicity Assessments of PolySia-NPs over 28 Days and the Potential Reversibility Following an Intravitreal (IVT) Injection to Dutch Belted Rabbits

Dutch Belted rabbits (8.5 months old) were administered a single intravitreal (IVT) injection of 50 µL/eye of PolySia-NPs at 1, 3, and 10 mg/mL (0.05, 0.15, and 0.5 mg/eye), using vehicle (10% sucrose) as a control. The animals were monitored for mortality and clinical signs, including body weights, body weight gains, food consumption, ophthalmology, intraocular pressure, changes in clinical pathology parameters (hematology, coagulation, and clinical chemistry), toxicokinetic parameters, aqueous humor collection, gross necropsy findings, organ weights, and histopathologic examinations. The potential toxicity was evaluated on Day 29 (male, N = 3 and female, N = 3), and the potential reversibility of any findings was assessed on Day 43 (male, N = 2 and female, N = 2). A complete gross pathological examination was performed on all main and recovery euthanasia animals, and organ weights were recorded. A detailed microscopic evaluation was performed on all tissues, including the eyes (target tissue) and optic nerves (selected tissue) from all recovery animals. For all main and recovery animals, an ocular histopathological examination was performed on at least 5 sagittal sections of each eye, with at least two sections containing the optic nerve (longitudinal) and one section containing the equivalent of the fovea/macula (i.e., the visual streak).

### 4.7. Toxicity Study at High Concentrations of PolySia-NPs Following an Intravitreal (IVT) Injection to Dutch Belted Rabbits

Briefly, animals were given a 50 μL IVT injection of ultra-high-dose (40 mg/mL) PolySia-NPs in the left eye (OS) and high-dose (10 mg/mL) PolySia-NPs in the right eye (OD) (N = 3/group). OEs and fundus imaging were performed at baseline and on Days 2, 4, and 7. Optical coherence tomography (OCT) was performed on Day 7 to measure outer nuclear layer (ONL) thickness. Ocular examinations (OEs) were performed at baseline and post-dose on Day 6. Inflammation was scored as follows: 0 = no inflammation; 1–4 = mild inflammation; 5–10 = moderate inflammation; and 11+ = severe inflammation.

### 4.8. In Vivo Toxicity Study in Non-Human Primates (NHPs) after Intravitreal (IVT) Administration

We selected non-human primates (NHPs) as a single species to evaluate the potential toxicology of PolySia-NPs due to their strong Siglec homology to humans compared to other species. To determine the potential toxicity of the PolySia-NPs given intravitreally once to cynomolgus monkeys at 0.05, 0.25, and 0.50 mg/eye into both eyes (OU), using 10% sucrose as the vehicle control group over 28 days (N = 3 females; N = 3 males), and to evaluate the potential reversibility of any findings at day 43 (N = 2 males; N = 2 females). The following parameters and endpoints were evaluated in this study: mortality, clinical signs, body weights, and body weight gains. Ophthalmological evaluations included indirect observations, ocular examinations (Oes) by slit-lamp and tonometry performed at baseline and on Days 1, 3, 7, and 14; also, intraocular pressure, optical coherence tomography (OCT), and electroretinography (ERG) readings were performed at baseline, on Days 14 and 28; and serum was collected at baseline, 1, 2, 4, 8, and 24 h post-injection, and on Days 3, 7, and 14 post-injections. Intraocular pressure was measured by rebound tonometry, and optical coherence tomography (OCT) imaging was performed by using the Heidelberg Spectralis HRA/OCT system for a single, horizontal, high-resolution line scan from the optic nerve head through the fovea. For ERG, animals were darkly adapted for a minimum of 1 h prior to readings. Full-field flash ERGs with Ganzfeld dome stimulus, with flash intensities according to ISCEV standard parameters and a light adaptation time of 5 min (Retiport Gamma, Roland Consult, or UTAS BigShot Visual Electrodiagnostic System); amplitude and latency values measured from tracings.

On Days 28 and 49 (following a 15-day recovery), animals were euthanized, and N = 2 eyes per group were collected and processed for histology.

## 5. Conclusions

PolySia-NPs were tested in vitro and in vivo for safety in both ocular intravitreal and systemic intravenous administrations. PolySia-NPs did not show in vitro cytotoxicity or mutagenesis capabilities. Intravenous (IV) administration of PolySia-NPs in mice did not result in any mortality or any effects on clinical signs, and the mammalian in vivo micronucleus test did not detect damage to the chromosomes or the mitotic apparatus of mouse erythroblasts. Similarly, respiratory system function in rats did not show any alteration after IV administration of PolySia-NPs.

Ocular safety examinations after intravitreal (IVT) administration of PolySia-NPs exhibited no adverse structural abnormalities or functional changes in rabbits or non-human primates (NHPs). Also supports a well-tolerated molecule in all safety pharmacological studies. We previously reported no safety concerns after IVT administration in mice [5] and are consistent with this result in larger mammals.

Based on the results of this study, intravitreal administration of PolySia-NPs is non-toxic and does not predict any potential toxicities when entering human administration. The results support a safe molecule for ocular delivery of PolySia-NPs in human clinical trials.

## Figures and Tables

**Figure 1 pharmaceuticals-17-00481-f001:**
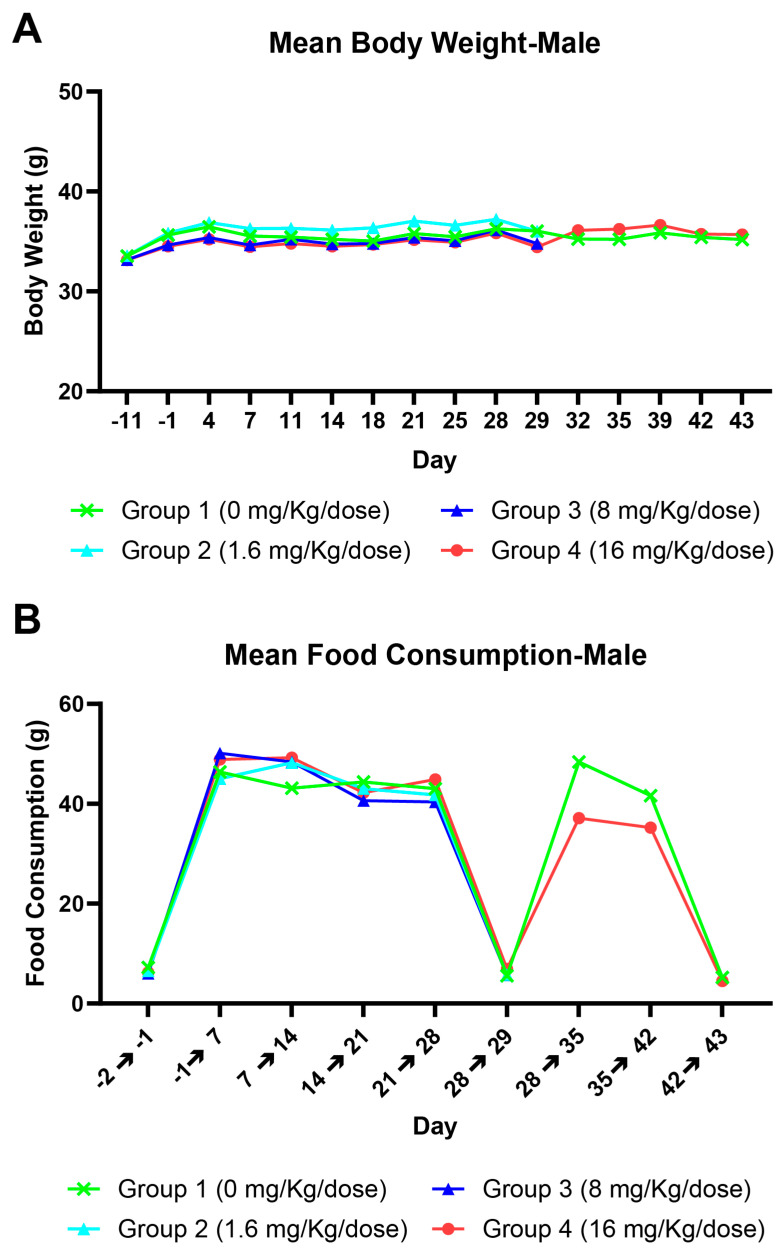
Relationship between PolySia-NPs vs. mean body weight (**A**) and mean food consumption (**B**) in mice.

**Figure 2 pharmaceuticals-17-00481-f002:**
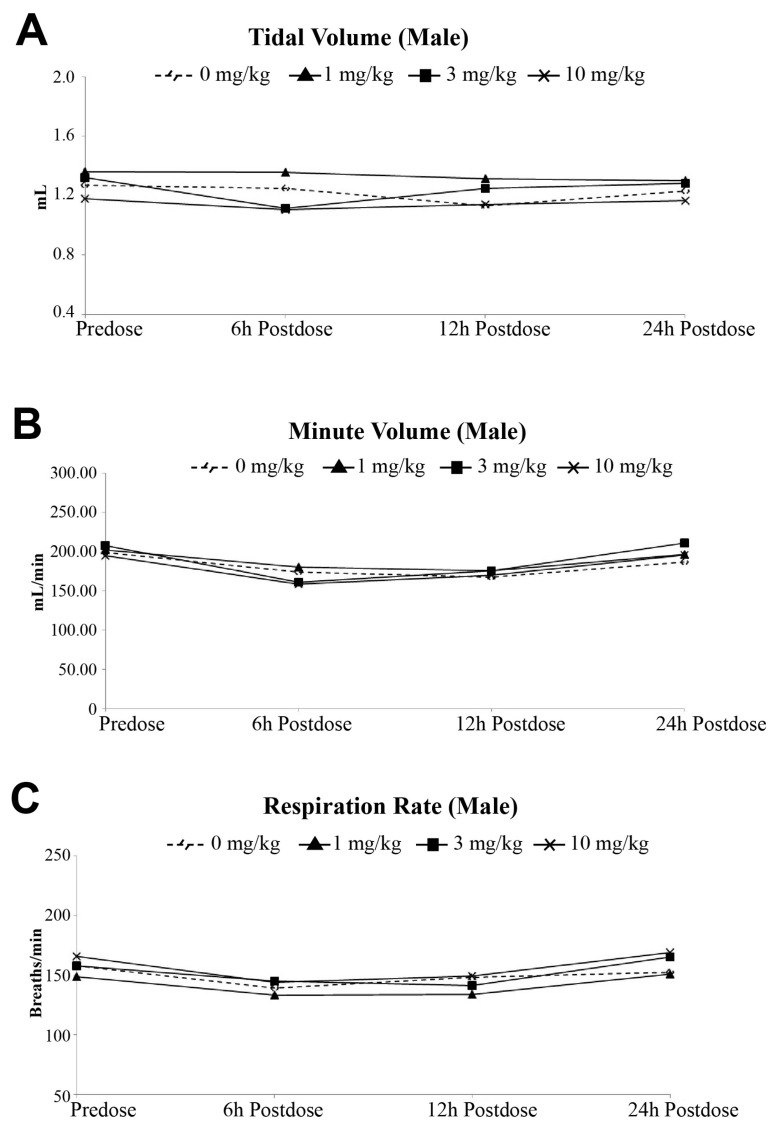
Effects of IV injection of PolySia-NPs on tidal volume (**A**), minute volume (**B**), and respiration rate (**C**) in Sprague Dawley rats.

**Figure 3 pharmaceuticals-17-00481-f003:**
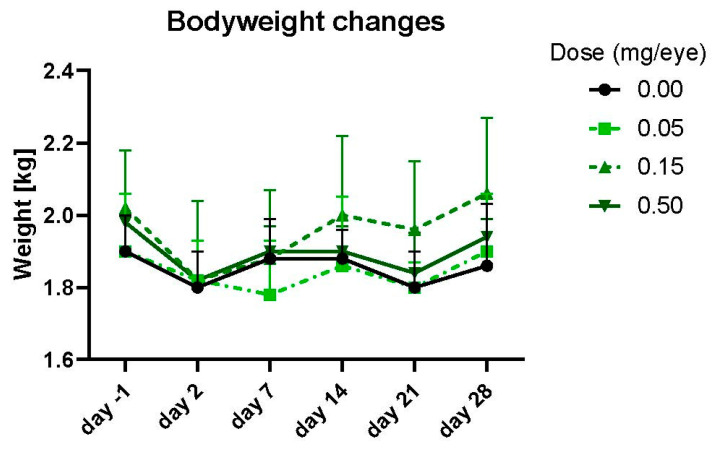
Effect of IVT administration of PolySia-NPs on body weight in Dutch Belted rabbits.

**Figure 4 pharmaceuticals-17-00481-f004:**
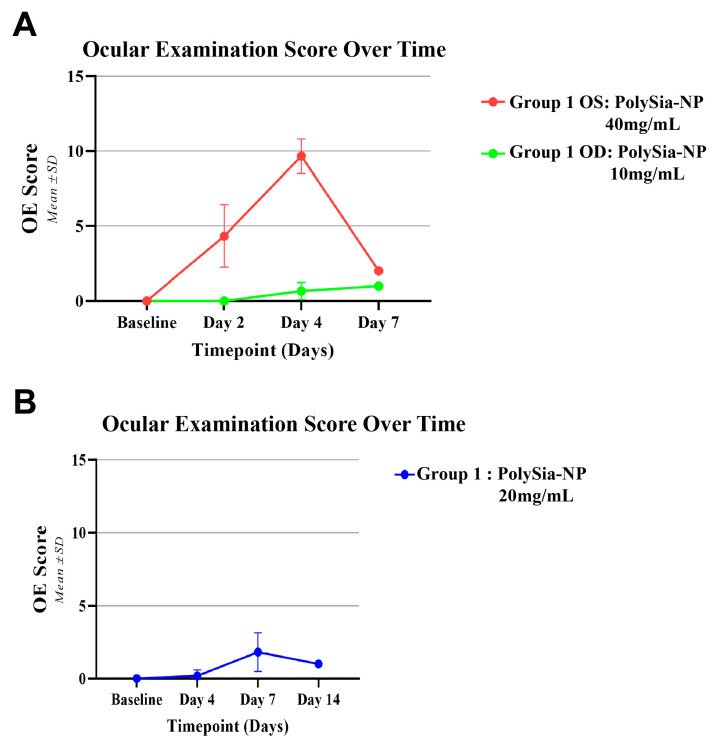
Ocular examination scores in OS and OD from rabbits treated with IVT injections of PolySia-NPs (**A**). OE score in rabbits treated with IVT injections of PolySia-NPs (**B**).

**Figure 5 pharmaceuticals-17-00481-f005:**
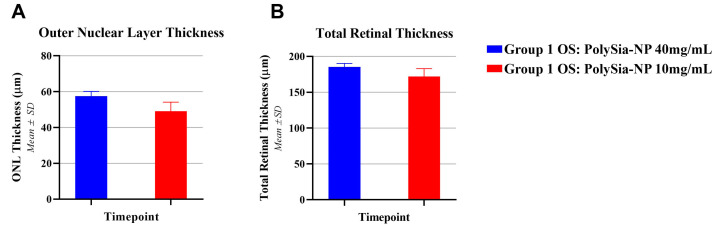
(**A**) ONL thickness and (**B**) total retinal thickness of the OS and OD groups.

**Figure 6 pharmaceuticals-17-00481-f006:**
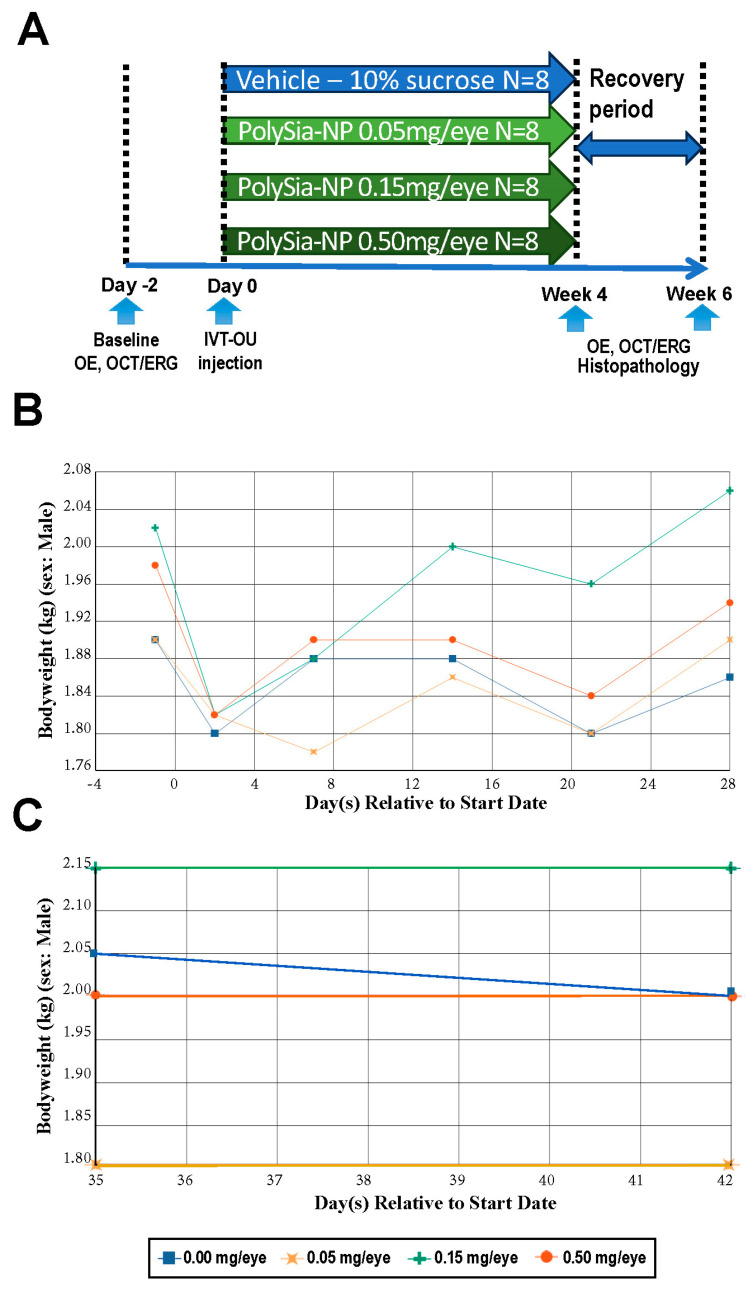
Experimental design (**A**) and bodyweight changes in male non-human primates after IVT administrations of PolySia-NPs recorded for 28 days (**B**) and from day 35 to day 42 (**C**).

**Figure 7 pharmaceuticals-17-00481-f007:**
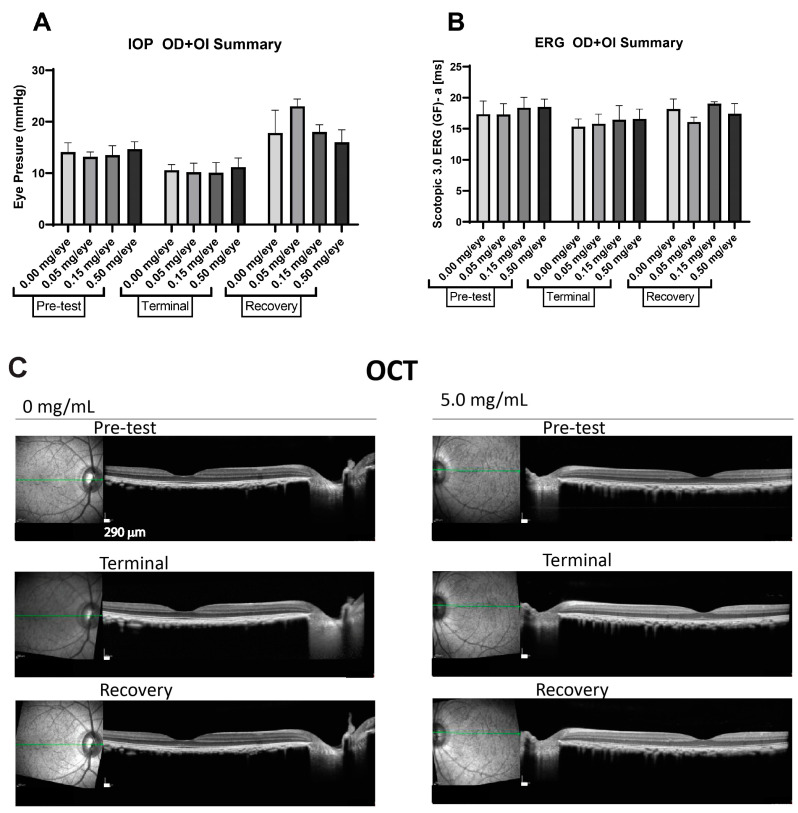
Effect of IVT administrations of PolySia-NPs on intraocular pressure (IOP) (**A**), electroretinography (ERG) (**B**), and optical coherence tomography (OCT) (**C**). Readings were performed at baseline (pre-test), on Day 14 (terminal), and on Day 28 (recovery). Green line represents the transversal axis of the eye for the OCT imaging.

**Table 1 pharmaceuticals-17-00481-t001:** Range-finding assay with and without metabolic activation to assess cytotoxicity.

Without Metabolic Activation						
Strain	Test Item	Dose Level per Plate	Mean Revertants per Plate	Standard Deviation	Ratio Treated	Individual Revertant Colony Counts
TA100	%10 Sucrose	100 µL	79.3	3.2	-	78,77,83
	PolySia-NP	1	97	0	1.2	97
		5	72	0	0.9	72
		10	69	0	0.9	69
		50	97	0	1.2	97
		100	84	0	1.1	84
		500	102	0	1.3	102
		1000	83	0	1	83
		5000	77	0	1	77
WP2uvrA	%10 Sucrose	100 µL	44.7	15	-	62,37,35
	PolySia-NP	1	44	0	1	44
		5	72	0	1.6	72
		10	51	0	1.1	51
		50	54	0	1.2	54
		100	61	0	1.4	61
		500	69	0	1.5	69
		1000	72	0	1.6	72
		5000	50	0	1.1	50
With Metabolic Activation						
Strain	Test Item	Dose Level per Plate	Mean Revertants per Plate	Standard Deviation	Ratio Treated	Individual Revertant Colony Counts
TA100	%10 Sucrose	100 µL	85	17.8	-	91,65,99
	PolySia-NP	1	90	0	1.1	90
		5	113	0	1.3	113
		10	76	0	0.9	76
		50	111	0	1.3	111
		100	70	0	0.8	70
		500	71	0	0.8	71
		1000	88	0	1	88
		5000	99	0	1.2	99
WP2uvrA	%10 Sucrose	100 µL	57.7	21.9	-	33,65,75
	PolySia-NP	1	64	0	1.1	64
		5	52	0	0.9	52
		10	54	0	0.9	54
		50	71	0	1.2	71
		100	43	0	0.7	43
		500	69	0	1.2	69
		1000	43	0	0.7	43
		5000	57	0	1	57

**Table 2 pharmaceuticals-17-00481-t002:** Mutagenesis assay with metabolic activation to assess cytotoxicity.

Mutagenesis Assay with Metabolic Activation						
Strain	Test Item	Dose Level per plate	Mean Revertants per Plate	Standard Deviation	Ratio Treated/Solvent	Individual Revertant Colony Counts
TA98	%10 Sucrose	61.3 µL	26	6.2	-	19,28,31
	PolySia-NP	31	25	3	1	25,22,28
		93	23	3.5	0.9	27,21,21
		278	26.7	1.5	1	25,27,28
		833	24.3	8.5	0.9	34,18,21
		2500	13.3	1.5	0.5	15,12,13
	2AA	2.5	1742	184.1	67	186,115,301,835
TA100	%10 Sucrose	61.3 µL	118.3	2.1	-	116,119,120
	PolySia-NP	31	90.3	6	0.8	91,84,96
		93	110.3	15	0.9	127,106,98
		278	93.7	11.7	0.8	81,96,104
		833	112.3	21.2	0.9	95,106,136
		2500	146	6.9	1.2	154,142,142
	2AA	2.5	1615.7	545.9	13.7	168,110,402,126
TA1535	%10 Sucrose	61.3 µL	9	2	-	11,9,7
	PolySia-NP	31	14.3	3.2	1.6	13,18,12
		93	12.3	1.5	1.4	11,14,12
		278	9	2.6	1	7,12,8
		833	12.7	2.9	1.4	11,16,11
		2500	13.3	2.1	1.5	14,11,15
	2AA	2.5	161.3	9.1	17.9	168,165,151
TA1537	%10 Sucrose	61.3 µL	8	4.6	-	7,13,4
	PolySia-NP	31	8.7	4.7	1.1	7,5,14
		93	6.7	2.5	0.8	9,7,4
		278	4.3	4.5	0.5	4,9,0
		833	8.3	3.5	1	12,5,8
		2500	6.7	5.1	0.8	8,11,1
	2AA	2.5	98.3	13.3	12.3	87,113,95
WP2uvrA	%10 Sucrose	61.3 µL	39	11.3	-	26,46,45
	PolySia-NP	31	37	9.2	0.9	27,39,45
		93	44.3	17.6	1.1	26,46,61
		278	49	8.7	1.3	55,53,39
		833	42.3	14.2	1.1	26,49,52
		2500	44.7	11.7	1.1	32,55,47
	2AA	10	396.7	136.4	10.2	500,448,242

Vehicle & Positive Controls. %10 sucrose, 2NF: 2-Nitrofluorene, ICR: ICR-191 Acridine, NQNO: 4-nitroquinoline-N-Oxide, SA: Sodium azide.

**Table 3 pharmaceuticals-17-00481-t003:** Mutagenesis assay without metabolic activation to assess cytotoxicity.

Mutagenesis Assay without Metabolic Activation						
Strain	Test Item	Dose Level per Plate	Mean Revertants per Plate	Standard Deviation	Ratio Treated/Solvent	Individual Revertant Colony Counts
TA98	%10 Sucrose	61.3 µL	22	4.4	-	20,27,19
	AVD-104 31		21	5	1	21,26,16
		93	15.7	4.7	0.7	12,21,14
		278	19.7	0.6	0.9	20,19,20
		833	8.3	2.5	0.4	6,11,8
		2500	15.3	2.5	0.7	13,18,15
	2NF	2.5	603.3	78.2	27.4	549,568,693
TA100	%10 Sucrose	61.3 µL	119	5	-	114,119,124
	AVD-104	31	104.7	16.8	0.9	94,96,124
		93	131.3	13.3	1.1	139,139,116
		278	127.7	11.1	1.1	138,116,129
		833	129.7	13.8	1.1	140,114,135
		2500	105.3	17.6	0.9	91,125,100
	SA	1	619.3	152.2	5.2	776,472,610
TA1535	%10 Sucrose	61.3 µL	11.7	0.6	-	12,12,11
	AVD-104	31	9.7	3.8	0.8	8,7,14
		93	9.3	5.1	0.8	5,8,15
		278	12	1	1	11,12,13
		833	7.7	1.5	0.7	9,6,8
		2500	9.7	1.2	0.8	9,9,11
	SA	1	639.7	17.6	54.8	656,621,642
TA1537	%10 Sucrose	61.3 µL	5.7	2.1	-	5,8,4
	AVD-104	31	3.3	2.1	0.6	1,5,4
		93	7	1	1.2	6,8,7
		278	10	1.7	1.8	12,9,9
		833	6.3	1.2	1.1	7,5,7
		2500	7	1	1.2	8,7,6
	ICR	0.5	118.3	6.7	20.9	126,115,114
WP2uvrA	%10 Sucrose	61.3 µL	29.3	9.5	-	40,22,26
	AVD-104	31	30.7	2.5	1	31,33,28
		93	30.7	11.2	1	35,18,39
		278	33	9	1.1	24,33,42
		833	30	10.1	1	21,28,41
		2500	42.3	14.8	1.4	26,55,46
	NQNO	2	907.3	85.6	30.9	929,980,813

Vehicle & Positive Controls. %10 sucrose, 2NF: 2-Nitrofluorene, ICR: ICR-191 Acridine, NQNO: 4-nitroquinoline-N-Oxide, SA: Sodium azide.

**Table 4 pharmaceuticals-17-00481-t004:** Micronucleus assay: four-hour treatment with metabolic activation to assess cytotoxicity.

Treatment (µg/mL)	Cytotoxicity RICC (%)	MN (%)	z′
10% Sucrose (2.6%)	0	0.89	NA
CP 4.7 µg/mL	50	5.58	0.79 *
CP 11.9 µg/mL	82	15.58	0.91 *
**PolySia-NP**			
**31.3**	−5	0.51	<0
**62.5**	−12	0.51	<0
**125**	−14	0.58	<0
**250**	−18	1.4	<0
**500**	3	0.3	0.24

MN: Micronucleated Cells, NA: Not applicable. CP: Cyclophosphamide Monohydrate. RICC: Relative Increase in cell count. * z′ ≥ 0.6.

**Table 5 pharmaceuticals-17-00481-t005:** Repeat micronucleus assay: four-hour treatment without metabolic activation to assess cytotoxicity.

Treatment (µg/mL)	Cytotoxicity RICC (%)	MN (%)	z′
10% Sucrose (2.6%)	0	0.3	NA
MMC 0.0625 µg/mL	50	3.47	0.78 *
MMC 0.125 µg/mL	63	8.7	0.88 *
**PolySia-NP**			
**31.3**	0	0.3	a
**62.5**	−10	0.26	<0
**125**	−2	0.42	<0
**250**	−23	0.39	<0
**500**	−4	0.35	<0

MN: Micronucleated Cells, NA: Not applicable. RICC: Relative Increase in cell count. MCC: Mitomycin C. * z′ ≥ 0.6. a Not calculated if Mean MN is the same as Vehicle Control.

**Table 6 pharmaceuticals-17-00481-t006:** Micronucleus assay: twenty-seven-hour treatment without metabolic activation.

Treatment (µg/mL)	Cytotoxicity RICC (%)	MN (%)	z′
10% Sucrose (2.6%)	0	0.37	NA
VIN 2.5 ng/mL	46	4.19	0.79 *
VIN 3.0 ng/mL	57	5.4	0.83 *
**PolySia-NP**			
**31.3**	−6	0.29	<0
**62.5**	−2	0.32	<0
**125**	−3	0.28	<0
**250**	3	0.36	<0
**500**	1	0.37	<0

MN: Micronucleated Cells. NA: Not applicable. VIN: Vinblastine Sulfate. * z′ ≥ 0.6. RICC: Relative Increase in cell count.

**Table 7 pharmaceuticals-17-00481-t007:** Definitive micronucleus assay to assess genotoxicity.

Sex: Male			
Treatment Group	Sampling Time (Hours)	PCE Percentage (Mean ± SD, %)	MN-PCE Frequency (Mean ± SD, %)
**Group 1**	1 (0.5–1.5)	56.3 ± 6.2	2.0 ± 0.8
Negative Control Article			
Dose			
(0 mg/kg/day)			
**Group 2**	1 (0.5–1.5)	58.9 ± 9.7	2.5 ± 0.4
AVD-104			
Dose			
(208.75 mg/kg/day)			
**Group 3**	1 (0.5–1.5)	56.9 ± 2.4	2.5 ± 0.3
AVD-104			
Dose			
(417.5 mg/kg/day)			
**Group 4**	1 (0.5–1.5)	57.4 ± 3.6	1.9 ± 0.7
AVD-104			
Dose			
(835 mg/kg/day)			
**Group 5**	22 (20~24)	47.7 ± 8.7	49.5 ± 16.6 *
(CP, 75 mg/kg)			

CP = cyclophosphamide monohydrate; * *p* ≤ 0.01.

**Table 8 pharmaceuticals-17-00481-t008:** Summary of microscopic findings during + terminal euthanasia and recovery + euthanasia.

Summary of Microscopic Findings–Terminal Euthanasia (Day 29)								
	Males				Females			
Group	1	2	3	4	1	2	3	4
Dose (mg/eye)	0	0.05	0.15	0.5	0	0.05	0.15	0.5
No. Animals per Group	3	3	3	3	3	3	3	3
Eyes (No. Examined)	3	3	3	3	3	3	3	3
Right: Infiltration, mononuclear cell, vitreous chamber	(0) a	0	2	3	0	0	2	3
Minimal Mild	0	0	2	1	0	0	2	1
Mild	0	0	0	2	0	0	0	2
Left: Infiltration, mononuclear cell, vitreous chamber	0	0	3	3	0	0	2	3
Minimal	0	0	3	1	0	0	2	2
Mild	0	0	0	2	0	0	0	1
^a^ Numbers in parentheses represent the number of animals with the finding								
**Summary of Microscopic Findings–Recovery Euthanasia (Day 43)**								
	Males				Females			
Group	1	2	3	4	1	2	3	4
Dose (mg/eye)	0	0.05	0.15	0.5	0	0.05	0.15	0.5
No. Animals per Group	2	2	2	2	2	2	2	2
Eyes (No. Examined)	2	2	2	2	2	2	2	2
Right: Infiltration, mononuclear cell, vitreous chamber	(0) a	0	1	2	0	0	1	2
Minimal Mild	0	0	1	1	0	0	1	1
Mild	0	0	0	1	0	0	0	1
Left: Infiltration, mononuclear cell, vitreous chamber	0	0	0	2	0	0	0	2
Minimal	0	0	0	1	0	0	0	2
Mild	0	0	0	1	0	0	0	0
^a^ Numbers in parentheses represent the number of animals with the finding								

## Data Availability

The raw data supporting the conclusions of this article will be made available by the authors on request.

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
