# Peer review of "Comprehensive Ocular and Systemic Safety Evaluation of Polysialic Acid-Decorated Immune Modulating Therapeutic Nanoparticles (PolySia-NPs) to Support Entry into First-in-Human Clinical Trials"

_pharmaceuticals, 2024, doi:10.3390/ph17040481_

Round 1

Reviewer 1 Report

Comments and Suggestions for Authors

The current manuscript aims to report that polysialic acid decorated immune modulating therapeutic nanoparticle (PolySia-NP) is safe and non-toxic systemically and in the eye. Although the topic is interesting in its scientific field, there are some issues that require the authors’ attention to improve the quality of this particular manuscript before further consideration for publication in a high-quality journal “Pharmaceuticals”.

Specific comments:

1.         As stated by the authors, animals received dose levels of 0, 167, 417.5 and 835 mg/kg/day from Groups 1 to 4 to determine the high-dose for the definitive micronucleus assay. However, the numerical values of dose in Group 2 was 208.75 mg/kg/day (Table 5A) rather than 167 mg/kg/day. Please carefully check the data presentation again and make necessary change.

2.         Furthermore, the results of Table 5A demonstrated abnormal clinical signs of skin discolored (tail, middle) in the 417.5 and 835 mg/kg/day dose groups. Nevertheless, the authors did not give any qualitative imaging data to support this important claim. Please improve.

3.         The data scale in the x-axis of Figure 1A/B is difficult to be understood and should be further specified.

4.         Please also clarify the underlying reason of the decrease of food consumption in mice during 7-10 days (Figure 1B). If the observation is linked with the influence of PolySia-NP, the authors should carefully check material biocompatibility again.

5.         The authors should specify the underlying reason of selecting different experimental time points for ocular examination in Figure 4A/B.

6.         As stated by the authors, the use of nanoparticles in pharmaceuticals has predominantly focused on delivering active drug substances. However, this important claim was not supported by any documented reference. In fact, many investigators have used inorganic nanoparticles in ocular delivery of pharmaceuticals (one example: DOI: 10.1002/advs.202302174). If possible, please consider the inclusion of this supportive case study in the reference list to enrich article content and balance scientific viewpoint.

Reviewer 2 Report

Comments and Suggestions for Authors

Major revision:

1.   Conclusions look too short for me

2.   Since the number of animals (3) look low, the authors need to add

extra paragraph to explain why do they consider it sufficient.

Comments on the Quality of English Language

It's well-written

Reviewer 3 Report

Comments and Suggestions for Authors

The paper submitted by Krishnan et al. deals with the in vivo administration and characterization of polysialic acid-functionalized PLGA nanoparticles. The ocular and systemic toxicity was further evaluated. The manuscript is clear and well written. However some correction are needed:

1. the title must be changed. Title cannot be a conclusion.

2. no data about the preparation of the nanoparticles was provided. The authors must describe the preparation method of these nanoparticles. 

3. line 96-97: a reference must be provided for the statement: "The use of nanoparticles in pharmaceuticals has predominantly focused on delivering active drug substances." A suggestion can be: https://doi.org/10.3390/polym16020206

4. as a general remark, there are too many tables. Some of them can be moved to Supporting Information.

5. for lines 166, 197, 217, 273: these statements are conclusions or titles of a sub-section? If there are titles, they cannot be formulated as a conclusion.

6. more detailed results must be provided in the conclusion section.

Round 2

Reviewer 1 Report

Comments and Suggestions for Authors

The revised version has adequately addressed most of the critiques raised by this reviewer and is now suitable for publication in "Pharmaceuticals".

Author Response

Thank you, we appreciate your feedbacks

Reviewer 2 Report

Comments and Suggestions for Authors

Panels B and C of Figure 6 are still too vague and the typeset is too small. The rest is fine

Author Response

Re: We added a paragraph with more details on the Figure 6 results. Panel B and C were enlarged and split in two with the supple. Fig. 1.